# Impact of midwife continuity of carer on stillbirth rate and first feed in England
Chris Roebuck [1,2] ✉, Jane Sandall [3], Robert West [4], Charlotte Atherden[2], Kate Parkyn[2] & Owen Johnson [1] ✉

## Abstract

**Background** In 2017 NHS England started rolling out a model where women have continuity of carer with the same midwifery team throughout the perinatal period. This study uses national data to test whether women of different groups receiving midwife continuity of carer had lower stillbirth rates and higher rates of a first feed of breast milk than women receiving standard care.

**Methods** We compared the two outcomes for women placed on the midwife continuity of carer pathway by 24 weeks and women receiving standard care in England, with logistic regression standardising between groups. We used the Maternity Services Dataset covering 922,149 women conceiving between 2020 and 2022.

**Results** Combining all demographic groups, women on a midwife continuity of carer pathway have a higher first feed of breast milk rate ($p < 0.001$), but do not show a difference in stillbirth rate, compared to women receiving standard care. However, Black women on this pathway have lower stillbirth rates ($p = 0.047$) compared to Black women receiving standard care, the only demographic group showing a difference. Women with no antenatal appointment at all by 24 weeks have much higher stillbirth rates than those with an appointment ($p < 0.001$).

**Conclusions** The findings that midwifery continuity of carer increases the first feed of breast milk uptake, which has health benefits. It may decrease stillbirth rates for Black women. Both findings inform future policy development and research. Further investigation and outreach around women not coming forward for timely antenatal care may also be beneficial.

## Plain language summary

In 2017, NHS England introduced continuity of midwife care, where women are supported by the same midwife team throughout pregnancy, birth, and after the baby's birth. We explored its impact using national data from women who became pregnant between 2020 and 2022. We compared women receiving this new approach of continuity of midwife care with those receiving standard care. More women under this new approach provided a first feed of breastmilk. Overall stillbirth rates were similar between the two groups. However, Black women under this new approach had lower stillbirth rates compared to Black women in standard care. We also found women with no antenatal appointments by 24 weeks of pregnancy were more likely to experience a stillbirth. These findings can help shape policies and research and encourage earlier access to antenatal services.

The best way of organising maternity services is the subject of a global debate, and there is evidence that midwife continuity of care teams throughout pregnancy and beyond can have positive outcomes[1]. In most developed countries, maternity care is provided by midwives who are members of multidisciplinary teams supporting women throughout pregnancy, during birth, and with the new infant. In the United Kingdom (UK), the National Health Service (NHS), provides free of charge, comprehensive maternity services, which are available to all. The management of the NHS is devolved to each of the four nations in the UK, and policies can vary between nations, with this study focusing on England. NHS England leads national healthcare policy, while NHS Trusts are the organisations that deliver care within specific geographical areas, including services provided in hospitals and the community.

In this study, the term 'women' is used to refer to pregnant individuals as this terminology is used in the national level, secondary data source which was utilised in this study. The authors acknowledge that this term refers to all individuals who may be pregnant, such as transgender and nonbinary individuals, who may not identify as women.

In England, pregnant women are encouraged to book their first antenatal appointment within 10 weeks of becoming pregnant, with 12 weeks considered late. This is typically followed by up to ten appointments with midwives, doctors, or other care professionals[2]. Childbirth most frequently happens in a hospital under the supervision of a midwife and doctor, if required, but can also take place in midwife-led units or at home, and for all women, this is followed by support after birth in the community by a midwife.[3–5] This model of maternity

[1]School of Computer Science, University of Leeds, Leeds, UK. [2]NHS England, Leeds, UK. [3]School of Life Course and Population Sciences, Faculty of Life Sciences & Medicine, King's College London, 10th Floor, North Wing, St. Thomas' Hospital, London, UK. [4]Leeds Institute of Health Sciences, University of Leeds, Leeds, UK. ✉e-mail: Chris.Roebuck@nhs.net; O.A.Johnson@leeds.ac.uk

care can involve the woman receiving care from multiple teams over this period.

A national maternity review was commissioned in 2015, with the resulting Better Births Policy[6], published in 2017, recommending the NHS should roll out a midwifery continuity of carer model (MCoC). An MCoC model provides dedicated support from the same midwifery team throughout pregnancy, birth, and the postnatal period[7]. It is intended to build stronger relationships, providing a safety net, enabling women to be listened to and be more comfortable raising concerns. Women are placed onto the MCoC model at an antenatal appointment before they reach 24 weeks of gestation.

In 2021, NHS England published the *Delivering MCoC at scale* policy, which presented the ambition for MCoC to be the default model of maternity care across all NHS Trusts in England[8]. In 2022, the national target timeframe was withdrawn due to staffing pressures[9] and shifted to an enhanced model of MCoC focused on women from Black, Asian, and minority ethnic groups and women from the most deprived groups as part of core20PLUS5[10], which is a major national initiative aiming to reduce healthcare inequalities. NHS Trusts decide where to place MCoC teams within their geographical area and consequently which women are placed on MCoC to best reduce inequalities.

Midwives have given positive qualitative feedback on MCoC, citing it helps women navigate the system and reach out more proactively if they have symptoms, and understand where to raise issues and that it can benefit childbearing women's mental health.[11,12] Research through randomised controlled trials (RCTs) shows important benefits in some but not all pregnancy outcomes. A Cochrane systematic review involving 18,533 women found that those who received MCoC were less likely to experience a caesarean section, instrumental birth, and episiotomy, and more likely to experience spontaneous vaginal birth and report more positive experiences during pregnancy, labour, and postpartum. The review highlighted there may be little or no difference in foetal loss after 24 weeks and neonatal death[1], but the size of the study may have been insufficient to detect differences in rare outcomes.

However, to our knowledge, no research has taken place on real-world implementation on a national dataset where increased sample size would have increased power to detect differences and examine specific population groups. This is a gap, as implementing an approach at scale across a population could have additional complexity that would influence the outcome. Larger national data allows testing on specific population sub-groups. NHS England has a mandate to collect and curate population-level cohort data. This includes the Maternity Services Data Set (MSDS), collecting person-level information in England covering a woman's journey from first appointment until discharge from maternity services, enabling the monitoring of service delivery, quality of care, and pregnancy outcomes. Launched in 2015, it has since undergone substantial development, gathering progressively richer and more complete data[13].

Pregnancy outcomes encompass the health of both the mother and the baby at the conclusion of the pregnancy, including live or stillbirth, miscarriage, neonatal outcomes such as first feed of breastmilk, and maternal outcomes such as post-partum haemorrhage. This study will focus on stillbirths and the neonatal outcome of the first feed of breastmilk.

Stillbirths are typically defined as where a baby is born dead after 24 completed weeks of pregnancy. They are a tragic outcome with global significance. While an estimated 98% of them occur in lower or middle-income countries[14], they remain a large concern in higher-income countries. In 2017, the British Government announced an ambition to halve the stillbirth rate from the 2010 baseline of 0.51% to 0.26% in England by 2025.[15,16] This ambition is a motivator for focusing on stillbirth as an outcome in this study.

Studies in the UK consistently show that the Black African and Caribbean population has the highest stillbirth rates of all main ethnic and demographic groups, and these rates have been found to be up to twice as high[17] as those for White women and remained higher after adjusting for other risk factors[18]. South Asian, particularly Pakistani and Bangladeshi women, and women living in more deprived areas also have higher rates

than their White or less deprived counterparts[17,18]. Qualitative reviews and reports showed that Black, Asian, and other minority groups often reported poor experiences of maternity services in the UK, including not being listened to, challenges in navigating the system, and a lack of trained interpreters for those who did not speak English[19]. Among those women who experienced a stillbirth or neonatal death, issues in the quality of care were more common in Black than White women, including barriers to accessing specific care[20]. More broadly, biological, clinical, behavioural, health service, and social factors contribute to different risks of stillbirth for different groups[14].

Another important pregnancy outcome, the first feed of breastmilk, is defined as a baby having their first feed consisting of maternal breastmilk or a donor's breastmilk. According to the World Health Organisation: early initiation of breastfeeding, within one hour of birth reduces newborn mortality[21]. This highlights its importance as a public health measure and supports its inclusion as an outcome in this study.

Breastfeeding rate also varies between ethnicities, with one study showing Black and other ethnic minority groups having higher breastfeeding rates than White women, thought to be driven by cultural expectations[22]. Another survey highlighted that breastfeeding decisions are mostly driven by knowledge, skills, and attitudes[23].

Separately, previous research has shown that women with a late first antenatal appointment were over-represented by more at-risk groups[24]. Studies from Finland and South Africa have shown women with a late first appointment had worse pregnancy outcomes, although the studies found no significant differences in stillbirth rate[25,26].

NHS England's access to rich, national data covering women's entire pregnancy, including the model of care they were placed on gave us a strong opportunity to investigate the effectiveness of the MCoC model on pregnancy outcomes. The aim of our study is to test in England:

- Hypothesis 1: women receiving MCoC had a) lower stillbirth rates and b) higher rates of a first feed of breast milk.
- Hypothesis 2: women who had no antenatal appointments by 24 weeks have a) higher stillbirth rates and b) lower rates of a first feed of breast milk compared to women seen by 24 weeks but not placed on MCoC pathway.

We test these hypotheses for all women and separately for each group targeted under the core20PLUS5 initiative: Black, Asian and women living in the most deprived areas. It is important to understand where MCoC could make the greatest impact to help future targeting decisions of the finite MCoC resource. It is possible that poorer interactions with maternity services over the course of a pregnancy would be a factor driving worse outcomes for higher-risk groups[27], alongside other clinical and broader health factors, and that MCoC could help mitigate these.

An additional objective is to test whether the Maternity Services Dataset can be made research-ready[28] to evaluate outcomes and, more broadly, whether analytical techniques applied to real-world data can be used in addition to RCTs to assess outcomes of interventions.

Our findings suggest that MCoC improves the uptake of the first feed of breast milk, which has health benefits. They also show it may have a positive impact on stillbirth rates for Black women, but do not indicate this for other demographic groups. They also suggest that women who do not have any antenatal appointments by 24 weeks have much higher stillbirth rates. These findings can inform future policy development and research, along with investigation and potential outreach to women who do not come forward for timely antenatal care.

## Methods
### Ethics approval and legal basis
All data used in the study were from the de-identified Maternity Services Dataset held by NHS Digital and NHS England, with no new data collection for this study. All data were analysed within the NHS Digital/ NHS England secure data environments.

Informed consent was not required because the study was a reuse of existing data that had direct identifiers removed, and the analysis was within the usage of the dataset that hospitals are requested to communicate to patients[29].

The NHS Digital Request for Analysis Process was followed, which provided ethical and Information Governance review, and had been established to review requests for NHS Digital employees to request access to data for analysis to benefit health and care.

Our request (reference IG-07865) was approved through this process on 9 September 2022. This approval required a senior requestor for the analysis (the then Clinical Director for Maternity Services within NHS England) and review and approval of the use of data by the Information Asset Owner for the Maternity Services Dataset, and the Director responsible for Data Access within NHS Digital.

The legal basis for the collection and use of the data is *The NHS Digital (Establishment of Information Systems for NHS Services: Maternity Services) Directions 2018*[30] transitioned to NHS England under the *NHS England De-Identified Data Analytics and Publication Directions 2023*[31].

Analysis of the data by ethnicity and the deprivation of the area in which the woman lived was central to our study. The Maternity Services Dataset does not currently capture information on gender identity so currently we are not able to report on this important area.

### Design and data preparation

To generate the MSDS, information is collected by clinicians and recorded by local maternity providers, who curate and submit it to NHS England. NHS England validates and links MSDS data across submissions where possible, and additional MSDS data fields (derivations) are calculated. Finally, MSDS data is de-identified, cleaned, and curated into tables for analysis (see Supplementary Fig. 1).

We used the pseudonymised unique pregnancy identifier to create a single record for each pregnancy covering all key characteristics of the woman, events, and outcomes. The data to generate the variables is shown in the Technical Output Specification[32]. This shows, for example, ethnicity was based on the 2001 census values[33]. The dataset often included multiple records for a given pregnancy. We developed rules creating a single value for each variable for each pregnancy, typically giving precedence to more definitive, more frequent, and more recent values.

Our study covered pregnancies with conceptions estimated to occur between October 2020 and December 2022. Within this period, there were 1,206,806 pregnancies in the Maternity Services Dataset. We chose this period as it had sufficient coverage, stable outcomes, and sufficient rollout of MCoC. Change in care delivery over the period due to COVID may have affected the models, but a shorter period would provide insufficient data. All 124 NHS Trusts that provided NHS-funded maternity services over the period had their data included.

We used estimated month of conception rather than the month of birth to avoid stillbirths, which have a different distribution of gestation length from live births, covering a different distribution of dates of the 24th week of pregnancy that would skew the analysis. We created a subset with singleton (only one child delivered) births, given the different outcome profile for non-singletons. This subset had recorded gestation period between 24 and 45 weeks, and those recorded as a live birth or stillbirth (excluding late abortions). These parameters excluded likely data issues, and the lower limit for gestation period aligned with the stillbirth definition. We removed data where the woman had an appointment by 24 weeks, but the MCoC placement status was unreported or incomplete. Finally, where the same woman had multiple sequential pregnancies in the study, we used the first pregnancy only to avoid these women having a larger impact on the findings. This left a total study population of 922,149 pregnancies.

We analysed two pre-specified primary outcomes: stillbirth and first feed of breast milk. These were selected based on their clinical significance, impact on longer-term health, clear definitions of what constitutes an outcome, and the completeness and clarity of the available data. For the first feed of breast milk, the dataset distinguishes whether the feed was from the mother or a donor—both of which we classified as breast milk—or whether the first feed was not breast milk. We assumed this information was recorded through direct observation. NHS Trusts are required to report these outcomes in the Maternity Services Dataset, which clearly indicates whether each event did or did not occur, with any unknowns distinctly highlighted. We did not consider other candidate outcomes, such as post-partum tears, as suitable to include in the study, as they did not have this clear differentiation in the data between the event not occurring or it simply not being recorded.

We defined a woman as placed on the full MCoC pathway by 24 weeks' pregnancy if the latest complete information stated she was on the pathway with a named lead midwife and care team by the final day of the 23rd week (167th day). We generated three cohorts of women comprising women whose latest status by 24 weeks showed they were placed on MCoC and met the full criteria, women whose latest status by 24 weeks showed they had an appointment but confirmed as not being placed on MCoC, and women with no appointment by 24 weeks.

Supplementary Fig. 2 shows the study inclusion flow diagram.

### Analysis

We built logistic regression models for each dependent variable: the two outcomes—stillbirth and first feed of breastmilk, and the cohorts to test the two hypotheses—whether the woman was placed on MCoC, and whether the woman had no appointment by 24 weeks. For some cases, the first feed of breast milk status was unknown, and these cases were excluded from the analysis of this outcome. Missing data for the independent variables were included as their own distinct category for that variable. Supplementary Table 1 presents the detailed approach to missing data and its rationale. We performed some sensitivity analysis to test the impact of alternative approaches to missing data in the independent variables. This sensitivity analysis compared the approach used in the study of missing values being treated as a distinct category in models with imputation and with complete case analysis.

To test the two hypotheses, the cohorts were also used as independent variables in the models predicting the two outcomes. We then created key demographic subsets and built separate logistic regression models for the two outcomes on these subsets. We did not adjust the significance threshold for the multiple outputs, as each one was considered a distinct hypothesis. Due to small numbers of stillbirths in each demographic subset, we used a less granular set of independent variables. Supplementary Data 1 shows the independent variables used in each multivariate model.

In all instances, we ran a univariable model, a multivariable model containing all independent variables except NHS Trust, and a multilevel model with all independent variables and NHS Trust as a random intercept. The size of the dataset meant most variables had a significant impact at $p < 0.001$ for the full cohorts of women not broken down by demographic groups, so for simplicity and consistency, we kept them all in for all models, rather than attempting to create more parsimonious models bespoke to each breakdown. Even though for some of the ethnic breakdowns, there were more variables without significant impact, we again favoured consistency in our models, so we included all the variables shown in Supplementary Data 1 rather than looking to remove variables that were not significant. The multilevel model was our preferred model, as in most, but not all, instances it performed best. It also led generally to more conservative outputs, decreasing the probability of other variables being highlighted as significant.

All independent variables were categorical and converted to dummy variables for each category. We presented the odds ratios between each variable value compared and the reference value for that variable, together with 95% confidence intervals. The z score was calculated as an interim output. The Wald test was used to transform this into the two-tailed p value that gave the level of significance in the difference of outcomes for each variable value compared with the reference value for that variable value. For all tests done on individual variable values, there was one degree of freedom. For completeness, our tables also presented an overall p-value for each

categorical variable (topic), for which the degrees of freedom were one less the number of variable values within that categorical variable.

We compared model performance on a given outcome using the akaike information criterion and for some of the key models we assessed the area under the receiver operating curve. We assumed a normal distribution of NHS Trust as the random intercept in the multilevel model, and while we did not formally test for normality, we used the intraclass correlation coefficient and for key models, we plotted caterpillar plots to understand in more depth the impact of NHS Trust as the random intercept. For each model, outcome, and independent variable combination, we reported the odds ratios and the crude rates. We placed 95% confidence intervals around both these using the Wilson methodology for binomial proportions.

Supplementary Note 1 gives detailed technical specifications.

### Reporting summary

Further information on research design is available in the Nature Portfolio Reporting Summary linked to this article.

## Results

### Women placed on pathway

We identified wide variation between Trusts in placing women on the MCoC pathway over the study period (Fig. 1). Of the 124 Trusts in the study, 30 were found to place fewer than 5% of women with known placement status on this pathway by 24 weeks of the pregnancy and four Trusts reported placing more than 95% of women with known placement status on this pathway. Supplementary Fig. 3 shows Trusts' MCoC placement rate by the average deprivation of residence of women seen at the Trust. Supplementary Table 2 shows a breakdown in placement rate by the region of the NHS Trust.

Supplementary Data 2 shows that the overall proportion of women placed on the MCoC pathway by 24 weeks of those with a known placement at this point was 23.1%. It shows that before and after standardisation for other factors such as age and ethnicity, significantly higher proportions of women were placed on the MCoC pathway who were younger (for example 27.0% under 20 s placed on MCoC compared with 22.8% 30–34 year olds), non-White British (for example 28.7% women from Caribbean ethnicity compared with 21.7% White British) and lived in more deprived areas (for example 25.3% in most deprived compared with 24.3% in least deprived). While statistically significant, the range of continuity of carer placement within these demographic groups is relatively small and, aside from groups with information unknown, range between 20 and 30%.

Supplementary Fig. 4 shows a national-level time series of MCoC placement status and other cohorts in the years preceding the study and the study period.

### Women with no appointment by 24 weeks

Supplementary Data 3 shows large and significant variation between demographic groups in the proportion of women with no appointment at all at 24 weeks. Women under 20, Black women of African origin, and women with no English address provided all have rates of no appointment by 24 weeks of over 10%. This compares with around 3% for the 30–34 year-old group, under 2% for the White British group, and 2% for the group living in the least deprived areas. After standardisation, the variation remains high but reduces slightly for women of Black African origin (adjusted odds ratio on multilevel model to White British of 9.90) and women with no English address provided (adjusted odds ratio on multilevel model to least deprived decile of 5.20).

### Stillbirth rate

Supplementary Data 4 shows that the overall stillbirth rate for the study population was 0.311%. It shows higher stillbirth rates for younger and older women compared to the 30–34 reference group, although after adjustment, only the 35–39 and 40–44 groups are significantly higher than the reference group at $p < 0.001$ under all models, with unadjusted stillbirth rates of 0.335% [0.310–0.363%] and 0.443% [0.384–0.511%] respectively. Stillbirth rates are significantly higher at $p < 0.001$ under all models for women with no English address provided (0.659% [0.488–0.889%]) and those living in the most deprived 10% of areas (0.438% [0.402–0.476%]) compared with the least deprived 10% (0.257% [0.222–0.298%]). They are also significantly higher under all models for women from Black and Asian ethnic groups, for example an unadjusted rate of 0.531% [0.463–0.610%] for women of Black African ethnicity and 0.470% [0.410–0.540%] for women of Pakistani ethnicity, compared with 0.274% [0.261–0.288%] for White British and 0.136% [0.069–0.268%] for women of Chinese ethnicity. From all the predictors, the group with the highest overall stillbirth rate was women who had a previous stillbirth, with an adjusted rate of 0.995% [0.852–1.160%].

For the whole study population, looking at hypothesis 1a, Supplementary Data 4 shows slightly lower stillbirth rates for women placed on MCoC at 24 weeks, compared to those not placed on MCoC, but this was not statistically significant under any model.

Looking at hypothesis 2a, women with no appointment by 24 weeks had significantly higher stillbirth rates. They had an unadjusted stillbirth rate of 0.721% and after adjusting for all variables in the study they had an odds ratio to women seen at 24 weeks but not placed on MCoC pathway of 2.161 [1.856–2.516], $p < 0.001$.

Looking at hypothesis 1a, Supplementary Data 5 shows that Black women placed on MCoC by 24 weeks pregnancy had lower stillbirth rates (0.402% [0.304–0.531%]) than Black women not placed on MCoC at this point (0.563% [0.491–0.646%]). Adjusting for all study variables, including NHS Trust, this was significant at $p < 0.05$ with an odds ratio 0.724 [0.526–0.995] in adjusted stillbirth rate between women placed and not placed on MCoC pathway. We found no significant variation in stillbirth rates by MCoC placement status for other ethnic or deprivation groups.

### Rate of first feed of breast milk

Supplementary Data 6 shows the overall rate of first feed of breastmilk was 72.7% [72.6–72.8%]. It shows significant differences by age under all models, ranging from an adjusted rate of 45.0% [44.3–45.7%] for under 20 s to 80.7% [80.3–81.1%] for 40–44-year-olds. White British women had the lowest rate of any ethnic group (66.3% [66.2–66.5%]) while women from other White backgrounds and women of Caribbean ethnicity had the highest rates (87.0% [86.7–87.2%] and 86.6% [85.8–87.3%] respectively). Rates of first feed of breast milk decreased with deprivation, with women living in the most deprived decile having a rate of 58.5% [58.2–58.8%], while those living in the least deprived decile had a rate of 83.0% [82.7–83.2%].

For the whole study population, looking at hypothesis 1b, Supplementary Data 6 shows significantly higher first feed of breast milk rates for women placed on the MCoC pathway at 24 weeks, compared to those seen by 24 weeks but not placed on the MCoC pathway, with a crude rate of 74.9% [74.7–75.1%] compared to 72.0% [71.9–72.1%]. Including NHS Trust in the statistical adjustment via the multilevel model reduced the effect size slightly, but it remained significant with an adjusted odds ratio of 1.072 [1.056–1.089] in first feed of breast milk rates between women placed and not placed on MCoC. Looking at hypothesis 2b, women with no appointment by 24 weeks had higher crude rates (73.5% [72.9–74.0%] but significantly lower standardised rates (odds ratio 0.907 [0.878–0.936] under multilevel model) of first feed of breastmilk compared to for those seen by 24 weeks and not placed on the MCoC pathway.

Supplementary Data 7 includes a breakdown of first feed of breastmilk for the cohorts by demographic group. It shows that White women [$p < 0.001$] and other ethnicities (covering mixed race, other and unknown) [$p = 0.002$] were the only ethnic groups with significant variation of first feed of breast milk by MCoC placement status, having adjusted for all variables under the multi-level model. Both these groups had higher rates for women placed on MCoC. It also shows that those women living in the most deprived quintile who were placed on MCoC had significantly higher rates of first feed of breast milk compared with those in this quintile who were not placed. Significantly higher rates of first feed of breast milk among women placed on MCoC compared with those not placed were also observed for the women living in the least deprived 80% of the population.

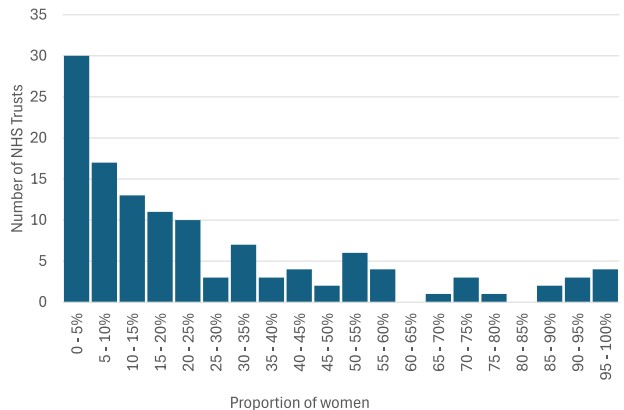

**Fig. 1 | Distribution of MCoC placement by NHS Trust.** Number of NHS Trusts with each range of proportions of women fully placed on MCoC by 24 weeks. The proportion of women fully placed on MCoC is derived from those with an appointment and known MCoC placement status by 24 weeks and covers conceptions occurring October 2020 – December 2022.

## Additional analysis

Supplementary Tables 3 to 5 show breakdowns of women in the study by pairs of key demographic variables. Supplementary Table 6 shows the results of the sensitivity analysis of different methods of handling missing data. Information around performance of the different models is in Supplementary Table 7 and, for the multilevel models, is plotted in Supplementary Fig. 5.

## Discussion

We used cohort data for around 900,000 childbearing women to test our hypotheses that firstly being placed on the MCoC pathway would lead to lower stillbirth rates and higher rates of first feed of breastmilk and secondly that not being seen at all by 24 weeks would lead to higher stillbirth rates and lower rates of first feed of breastmilk. We examined these for all women and for each demographic group with high risk factors who are targeted under the core20PLUS5 initiative (ethnicity and area of deprivation).

Looking at our first hypothesis for all women together, there was no significant difference in the rate of stillbirth for women placed on the MCoC pathway compared to those not placed on the pathway. This is consistent with no significant difference being found in the equivalent foetal loss measure in the previous systematic review[1].

However, examining the groups with high risk factors separately, we found that Black women had significantly lower rates of stillbirth when placed on the full MCoC pathway compared with Black women not placed on the pathway. We found no significant differences in stillbirths by MCoC group for women living in the most deprived quintile of areas, nor for women of South Asian ethnicity. Neither were significant differences found for the broad groups outside core20PLUS5, comprising White women and women not living in the most deprived quintile. Black women have higher stillbirth rates than other ethnic or deprivation category[18] and this study provides some evidence that MCoC can help reduce their stillbirth rates. To our knowledge, no previous studies have investigated this.

Some caution is needed, given Black women were the only group with significant differences and with relatively small numbers leading to higher levels of potential chance variation. While the effect size was large (adjusted odds ratio 0.724 [0.526–0.995]), it was just significant ($p = 0.047$). Each group was tested as a separate hypothesis, increasing the possibility of a single positive result due to chance. Given the specific characteristics of each demographic group, this risk was deemed preferable to that of potentially missing a significant finding.

Further research on why Black women appear to benefit more than other groups would be beneficial and could look by ethnic group at the differences in type, timeliness and frequency of interactions with maternity services for women on and not on the MCoC pathway, together with

qualitative feedback around levels of access and support. The variety of factors behind stillbirth and their potential to be mediated through MCoC could be examined in more depth, for example, some South Asian communities may have more stillbirths due to cultural differences around terminating pregnancies[18] on which MCoC may not impact.

Women placed on the MCoC pathway had a significantly higher rate of first feed of breast milk than those not placed on the pathway, with a 3 percentage point difference in rates. While the positive impact of being placed on the MCoC pathway remained significant after adjusting for all variables in the study, the size of the impact did reduce slightly once NHS Trust was adjusted for in the multilevel model, demonstrating that NHS Trust also had an important role. This differs from the findings in the systematic review[1] that stated there may be little or no difference in breastfeeding initiation, highlighting low certainty of the evidence. Our study involved around 100 times more women than the pooled randomised control trials from the systematic review, so we had increased power to detect differences.

First feed of breast milk is a much more common outcome than stillbirth, so for first feed of breast milk, a smaller change leads to much more significant results. However, of the two outcomes, stillbirth has a far greater impact, so the fact that for Black women, MCoC placement has a significant impact on stillbirth outcomes is potentially a more important finding.

The overall proportion of 23% of women placed on MCoC broadly met expectations given the ambition outlined in 2021 for MCoC to be the default model of care, but a further announcement in 2022 made the implementation timeline dependent on staff capacity. The relatively even distribution by demographic group, but wider variation by NHS Trust and Region, suggests differing prioritisation between regions and organisations, but Supplementary Fig. 3 shows this is not related to deprivation of the areas they serve. Lower variation in MCoC placement by other demographic groups suggests that within organisations, there has generally been little targeting by demographic group, but this is likely to change under the core20PLUS5 policy.

Looking at broader implications on MCoC within England, the findings can help inform decision making on the rollout of enhanced MCoC, targeted specifically at deprived areas and ethnic minority groups as part of the core20PLUS5, suggesting that for Black women it may have a positive impact on stillbirth rate, and for all women improve the uptake of first feed of breast milk, which has health benefits[21]. As the rollout progresses, outcomes should be monitored to see if these patterns continue. Further research has been commissioned by the National Institute for Health Research to examine the implementation of the enhanced MCoC model[34].

Many of the study's findings on MCoC should be generalisable to other developed countries where the same organisation is responsible for a woman's prenatal care, labour, and postnatal care, as this structure would enable MCoC implementation. Trials of MCoC have operated in Australia, Ireland, Canada and China[1]. With an ongoing trial looking at the impact on preterm birth among vulnerable and disadvantaged women[35]. Tailored implementation would be required based on the country's precise system. Findings on disadvantaged groups are likely to be broadly transferable. For demographic comparisons, however, context will be particularly important. Caution should be applied in generalising the findings on ethnicity to other countries. The local context on why and when immigration took place and levels of integration are likely to affect outcomes. There are likely to be important differences within broad ethnic groups based on this context, and we have presented the most granular available ethnicity data in our descriptive tables, for example, separating Pakistani and Bangladeshi ethnicities and separating Black African and Black Caribbean.

For our second hypothesis on women who had no antenatal appointment by 24 weeks, we found larger differences, with this group having over double the crude stillbirth rate of those who had an appointment. Women with no appointment had higher representation from high-risk groups, with much higher rates in the African ethnic group and among 20-year-olds. This is consistent with other studies, which show higher rates of late appointments for more at-risk groups[27]. However, even after

adjustment for these factors, the odds ratio was over 2 for stillbirths between those women with no appointment and those seen but not placed on MCoC. They also had slightly lower rates of first feed of breast milk once adjustment for the factors had occurred.

This stillbirth finding contrasts with the South African[25] and Finnish[26] studies, which found no significant differences in stillbirth rates for women seen very late. In the South African study, this may be due to the service organisation, as around a third of women did not have their first appointment until the 27th week of pregnancy or later, compared to under 3% after 24 weeks in our study. The South African study also excluded preterm stillbirths. The Finnish study had a lower cutoff of 16 weeks to define a very late first appointment, which could reduce any impact. It found more adverse pregnancy outcomes, such as preterm birth for very late first appointments, but not for perinatal mortality (stillbirths and deaths in the first week of birth), although low numbers may have impacted.

Further investigation is important on whether the worse outcomes for late engagers in England are directly due to this late engagement through factors including fewer appointments and later scans and tests, limiting detection of problems, or indirect links. These could include wider behavioural patterns separately causing late attendances and increased stillbirth risk. The difference in findings between this and the international studies may mean that the generalisability of these results is limited to developed nations, where most women have their first antenatal appointment in the first trimester. It may also be dependent on the characteristics of women who have late appointments in different countries. Service providers may wish to focus on encouraging a timely uptake of available services by all women, with a particular focus on those groups who are currently most likely to have a late or very late first appointment.

Our study has several strengths. To our knowledge, it is the largest study assessing the impact of MCoC, using an entire nation's real-world data with all associated diversity and complexity. The size and timeliness of data in Maternity Services Dataset enables rare events to be studied even for population subgroups. Its information depth, including from antenatal appointments, enables a rich picture of factors affecting key outcomes to be built and monthly updates with rapid information on changes to these factors.

The biggest limitation, like the biggest strength, is the real-world data. The comparison between the non-randomly allocated cohorts may not be totally fair despite the many steps we took to address this. There would be local variation in the recording and implementation of MCoC. Data on the care women received is not readily available, so the analysis was on the care pathway the women were placed on and in some instances, they may have moved onto a different pathway, which was not captured. It would be beneficial for subsequent versions of MSDS to capture whether MCoC was received, as well as whether the woman was placed on the pathway. There may be systematic differences between cohorts not captured in the data items or complex interactions between variables in the data items not picked up in the models. There may be unknown factors that could affect both the cohort and outcomes. RCTs would be less impacted by these limitations. The predictive power of most models shown in Supplementary Table 7 is relatively modest, so there may be some residual systemic between-group differences not accounted for.

Missing and unknown values in the data raised methodological questions, and our approach is fully set out in Supplementary Table 1. We viewed it as a clear-cut decision only to include women who were specifically recorded as having been placed or not placed on MCoC by 24 weeks, or the woman had no appointment by 24 weeks. Only those with a recorded outcome could be included in the analysis of that outcome. The approach to missing or unknown values in some independent variables that ranged from 2 to 4% was less clear-cut. We considered full case analysis, but had concerns that removals could bias the data. We considered imputation but given the variety of factors causing missing data, it was not clear what assumptions to include. Therefore, we included the missing data flag for a given variable as a distinct category for that variable in the regression model, which has been validated as a possible approach in various settings[36]. Most importantly,

sensitivity analysis in Supplementary Table 6 shows the main study findings are preserved under all three approaches.

To conclude, our study has shown that women placed on the Midwife Continuity of Carer pathway had higher rates of a first feed of breast milk and Black women placed on the Midwife Continuity of Carer pathway had lower stillbirth rates. It also showed that women who had no antenatal appointment at all by 24 weeks had much higher stillbirth rates compared to those who did have an appointment under any care pathway. It demonstrated the strength of large-scale real-world clinical data in being able to explore the impact of MCoC on different ethnic and demographic groups, which smaller RCTs and meta-analyses had not enabled. It acknowledged the risk that, despite adjusting for relevant factors to make the cohorts as comparable as possible, there may be some residual differences between them that would not be present in a randomised trial, and it highlighted areas for further investigation.

## Data availability
All aggregate data supporting the findings of this study are available within the paper, its Supplementary Information and Supplementary Data. The source data underlying the Fig. 1 and Supplementary Fig. 3, 4 and 5 can be found in Supplementary Data 8 – 10. Other aggregated Maternity Services Data relating to this study are available from the corresponding authors on reasonable request. The underlying record level data used in this research, including both the Maternity Services Dataset (MSDS) and the curated record level outcome file that fed the logistic regression models are de-identified but at patient record level, so a legal basis is required to obtain access. Access to the MSDS can be requested through NHS England's Data Access Request Service (DARS) https://digital.nhs.uk/services/data-access-request-service-dars and made available to requesters who meet the legal requirements. This is a chargeable service to cover the cost of running it.

## Code availability
The code is available in an online repository[37]. This code uses standard data processing techniques and statistical tests and is made available for transparency. As the data pipeline starts in NHS England's secure Data Access Environment, it requires access to this or access to the underlying data through NHS England's Data Access Request Service (DARS) to run it (see details in data availability section). Our logistic univariable and multi-variable regression models were run on R version 4.2.1 using finalfit (1.0.7). The multilevel models used the same packages but finalfit utilised the lmer function lme4 (1.1-35.1). The first part of the data pipeline was built on NHS England's data access environment using version 3.68 of Databricks. Within Databricks, we created a cleaned and curated set of definitive variables for each pregnancy using SQL alongside version 1.21.2 of Pandas within Python. The statistical modelling on these variables to produce the final outputs was performed in R-studio (R version 4.2.1) on NHS England's remote desktop services. It employed the following R packages: Matrix (1.6-5), lme4 (1.1-35.1), dplyr (1.1.4), finalfit (1.0.7), openxlsx (4.2.5.2), Desc-Tools (0.99.54), mlmhelpr (0.1.0), pROC (1.18.5). A parallel run of some of the statistical modelling was done using Scikit-learn within Python to check it generated similar numbers to the R output as an additional layer of quality assurance.

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

## Acknowledgements

The first draft of this research was produced as part of Chris Roebuck's Master of Research in Data Science and Health Analytics at the University of Leeds. Chris Roebuck was supported in part as an aligned member of the Leeds UKRI CDT in AI for Medical Diagnosis and Care (UKRI grant number EP/S024336/1). We would like to acknowledge NHS England (previously NHS Digital) for their support throughout this research. This research was then developed with the co-authors following this course concluding. There

was no research grant. JS is an NIHR Senior Investigator Emeritus and is supported by the National Institute for Health Research (NIHR) Applied Research Collaboration South London (NIHR ARC South London) at King's College Hospital NHS Foundation Trust. The views expressed are those of the authors and not necessarily those of the NIHR, the Department of Health and Social Care or NHS England.

## Author contributions

J.S., O.J., C.R. and K.P. conceptualised the project. C.R., R.W. and K.P. designed the methodology. C.R. and C.A. coded the data and analytical pipelines. C.A., K.P. and C.R. provided validation of the outputs. C.R. performed the statistical analysis and investigation, with guidance from R.W., K.P. and C.R. curated the data. C.R. wrote the original draft. J.S., O.J., R.W., C.A. and K.P. provided review and editing. C.R. provided visualisations. J.S., O.J. and R.W. provided supervision.

## Competing interests

The authors declare the following competing interests. JS was seconded to NHS England February 2021 to August 2023.CR, CH and KP are employed by NHS England, a public body at Arm's Length from Government responsible for managing the NHS in England. All other authors declare they have no competing interests.
