## [Transparent Peer Review file · Communications Medicine]

Impact of midwife continuity of carer on stillbirth rate and first feed in England

Corresponding Author: Mr Chris Roebuck

Version 0:

Reviewer comments:

Reviewer #1

(Remarks to the Author)

Thank you very much for the opportunity to review this very interesting manuscript. This presents an interesting analysis of administrative data at a National level and presents interesting findings with the potential to improve care and maternal health equity, especially for those most marginalized. While the approach and the findings are interesting, I offer some points that could improve the manuscript below.

Looking at the data, which is fascinating and I feel like there is a lot of rich research that could come out of the MSDS, I am left wondering about a number of things:

1. How many of the characteristics intersect (i.e. Black, higher deprivation, young)?
 2. Also, are the trusts that place women in MCoC located in areas with higher social vulnerability?
 3. How are women generally placed on MCoC? Are they assigned? Can women request it?
 4. Would someone want to request it or are there drawbacks such as long waiting times or less desirable clinic locations?
- Some more information might be helpful for non-English readers to understand the context of MCoC in the NHS.

Results

These tables are very interesting. I understand that you have beautiful granularity of data but I wonder if collapsing some of the categories might make the data more impactful? For example, I see that you have deciles of deprivation, it might make more meaningful and clinical sense to convert them to quartiles or quintiles. Similarly with race/ethnicity. I am not sure that the categories White and Black Caribbean vs White and Black African, while certainly meaningful to those individuals, might not be meaningfully different to readers or to clinicians caring for individuals from these groups. As I do not live in England, I do not know if this holds true in your context. The detailed tables could be included as supplementary with simpler tables included in the main portion of the article.

Overall, a very good exploration of fascinating data. I would just like a bit more synthesis and also, discussion of the implications of the findings.

Also, I am not sure why the Methods section is at the end of the manuscript rather than before Results. Further, while the methods and analysis sections are extremely important to determine the rigor of a study, the multiple pages of Methods and Analysis are a bit long and too detailed for this descriptive study.

Some specific comments to improve the writing are included below:

This is a confusing sentence as written, especially with inclusion of the word "delivery" which, for some readers, may mean "birth":

L48 In the United Kingdom (UK) National Health Service (NHS), the full range of maternity services are provided by NHS Trusts free at the point of delivery.

In the first paragraph you twice state that midwifery continuity of care may have better outcomes, but you do not include any citations in either instance.

This feels a bit redundant as written:

L62 NHS England is the national body that leads the NHS in England

Need a citation for these statements:

L76 these rates have been found 77 to be up to twice as high as those for white women (CITE) and were still higher after adjusting for other 78 risk factors (CITE)

L87-90: You speak to negative experiences with maternity carers and their contribution to higher stillbirth rates. While this is likely true, you need to connect the dots a bit more here.

Reviewer #2

(Remarks to the Author)

Thank you for the opportunity to review the manuscript.

This manuscript presents a study that utilized data from approximately 900,000 childbearing women in England to evaluate the impact of midwifery continuity of carer (MCoC) on two birth outcomes, specifically stillbirth rates and initial breastfeeding rates. The study notably examines these outcomes across demographic groups with high-risk factors identified under the core20PLUS5 initiative, additionally focusing on women who had not been seen in any care pathway by 24 weeks gestation. This work is based on extensive cohort data collected and administered by NHS England, providing insights into a real-world implementation of the midwifery-led continuity of care model.

Strengths of the Study

The dataset employed is uniquely large and allows for meaningful analysis across various sociodemographic groups, making the study highly relevant both within and outside the UK. By using a population-based dataset, the manuscript addresses a critical public health goal, aligning with government initiatives aimed at reducing stillbirth rates in the UK. The study contributes valuable evidence supporting MCoC's effectiveness, especially among Black, as well as in deprived populations, aiming to close healthcare access gaps.

Title and Scope

The title, "Closing the Inequalities Gap: Impact of Midwife Continuity of Carer on Birth Outcomes," appears broader than the study's scope, which focuses primarily on stillbirth and breastfeeding initiation rates. Although the title highlights "birth outcomes," specifying the exact variables studied (e.g., stillbirth rates and first feed of breastmilk) would align it more closely with the manuscript's content. This change would provide clearer expectations to the reader and align better with the study's focus on minority and deprived populations.

Introduction and Study Goals

The introduction offers an overview of MCoC and addresses its role in addressing inequalities in birth outcomes. However, the section could benefit from reorganization to enhance clarity and flow. For instance, the context of perinatal care and the structure of MCoC in England should be introduced first, followed by a description of access disparities faced by minority groups and the overview of the study outcomes. The current structure of this section briefly mentions the two outcomes at first but without providing the rationale for their selection, making it unclear why these variables were analyzed together. Since they do not belong to the same category, this raises questions about the criteria used for selecting variables for analysis. Reorganizing the introduction would give readers essential context for understanding the choice of outcomes and independent variables examined.

Lines 115-119 present the research objective described in the manuscript. The general aim stated at the beginning of the introduction appears somewhat misleading, thus perhaps it shouldn't be mentioned at this point and in such a general way. The study goals and hypotheses should also be placed together in one section along with their rationale at the end of the introduction section. It would be useful to pair the research objectives with the hypotheses (perhaps numbered) to make it easier to locate and reference them during the analysis of results.

The line, "We avoided outcomes with poorer data quality or where clinical context of individual cases would be needed," (lines 118-119) is somewhat vague. Providing examples of outcomes that were excluded due to data limitations would clarify the study's design and limitations.

Methods

The Methods section, currently located after the Discussion, would ideally follow the Introduction to provide a logical flow. This section is otherwise clear in outlining the study's inclusion and exclusion criteria, group classifications, and outcome measures. However, the variable "first feed of breastmilk" lacks clarity in terms of its measurement. Details on whether breastfeeding was directly observed or inferred based on database entries would strengthen this section.

Additionally, "NHS Trust" is mentioned in both the Introduction, Methods and Results, yet many readers outside the UK may be unfamiliar with this term. Defining "NHS Trust" early on would aid comprehension.

Line 135: It might be helpful to repeat the full name, "Maternity Services Data Set," next to the abbreviation MSDS to remind readers what it stands for.

Results

The Results section is well-structured, with clear tables that present the data effectively. In line 151, the text references 121 NHS Trusts with more than 1,000 deliveries. Clarifying whether data from only these Trusts were used, and how this subset relates to the total number of Trusts in the UK, would enhance interpretation.

Supplementary file attached to the review includes my comments on the supplementary material.

Discussion

The discussion section covers the most important aspects of the results; however, expanding it could benefit readers. For instance, several points caught my attention: Is the 20-30% placement of women in the MCoC what the authors expected? Do the results reveal any specific characteristics of the trusts with low vs. high MCoC placement rates, such as their location in more or less socially and economically deprived districts? How do the authors interpret the characteristics of the groups placed in the MCoC—is this distribution in line with expectations? Additionally, why did placement in MCoC have a relatively low impact on stillbirth rates, while the effect on the first breastmilk feed was more noticeable?

The paragraph discussing the higher stillbirth rates among women with no antenatal care before 24 weeks is particularly informative.

Language and Style Issues Several minor language issues affect clarity:

- Line 125: "If this, is the case..." should remove the comma.
- Line 133: "they had some worse birth outcomes..." could benefit from more formal wording, instead of the word 'some'.
- Line 145: "not on placed on the MCoC" should be corrected to "not placed on the MCoC."
- Line 394-395: "were selected" could be omitted.
- Line 409: "was the total population used any aspects of the study." appears incomplete.
- Line 468: "Table S5" should be updated to "figures S5a-S5e."

Conclusion

Overall, this manuscript represents a substantial contribution to the evidence on MCoC in reducing health disparities among minority and deprived populations. The study's significant sample size and real-world relevance underscore its value. With some reorganization, clearer hypotheses, and minor language adjustments, the manuscript would provide a highly impactful account of the role of MCoC in supporting at-risk populations within the healthcare system, not only in UK.

I recommend the presented manuscript for publication after implementing the proposed changes. Thank you again for the opportunity to review this very interesting and important manuscript.

Paulina Pawlicka, PhD

Reviewer #3

(Remarks to the Author)

Thank you for the opportunity to review this manuscript. I appreciate the importance of the continuity of carer model and believe this is important topic. I have several major concerns about manuscript submitted for publication. I think the manuscript can be written in a way that is clearer for readers and suggest that there be some reorganization especially in the introduction.

1. For your introduction (i.e., the main), this could be set up clearer for readers. It is unclear from the introduction what the gap is trying to fill with this article. How is this paper different from the other continuity of carer articles out there including the systematic reviews on continuity of carer? I do appreciate the discussion of NHS England, but this should be discussed together. There should also be sentence about what is the aim of this study and your hypotheses which were described in line 115 and 120. I'm also confused why there was some discussion of the methods in this. The introduction could be setup in clearer way for readers.

2. A study flow diagram would be helpful instead of the Table S1. I suggest putting this in the traditional study flow diagram. In addition, why were there two different sample sizes?

3. What are the "Trusts" referred to in line 150-151, etc.?

4. Similarly with the discussion, there really is not a discussion of how their findings are similar or different from previous literature or what is known, which can be done with this study. I think this would make the paper stronger than discussing the results in further in-depth. In addition, I appreciate a discussion regarding the implications for practice and/or policy.

5. In the discussion, you mention that there was a missing data. What was the percentage of people who missing data? Did you do complete case analysis or imputation? In the methods section, you should include a description about how you handled this. I would like to understand more about the missing data because you mentioned these people had worse outcomes.

6. What about the generalizability of your study findings?

7. Given the complex nature of race/ethnicity, I think there should be a discussion about this in your manuscript. Is it similar

to the United States?

8. I suggest using gender neutral terminology.

Version 1:

Reviewer comments:

Reviewer #1

(Remarks to the Author)

Thank you for your careful attention to the many reviewer comments. The revised manuscript is stronger and brings significantly more clarity. Well done. I look forward to seeing this published soon.

Reviewer #2

(Remarks to the Author)

Dear Authors,

Thank You for addressing all the amendments to the manuscript. All of my concerns and comments have been thoroughly addressed, and the changes have been thoroughly marked and explained. The manuscript now reads very well, and is concise and coherent. I have no further comments or requests and recommend the manuscript for publication.

To: reviewers of *“Closing the inequalities gap: impact of midwife continuity of carer on stillbirth rate and first feed of breast milk, an English population-based cohort study”*

c/o Dr Lauren Malave

Associate Editor

Communications Medicine

Dear Reviewers

We would like to thank you very much for taking the time to produce such a helpful, actionable and thorough set of feedback on our draft manuscript. We have made some changes to the manuscript and supplementary information in response to your comments. Please see enclosed a table that outlines our response to each individual item you highlighted. We hope we have addressed each point to your satisfaction, and we look forward to any further feedback you have.

Yours faithfully

Chris Roebuck

Chris.roebuck@nhs.net

Deputy Director of Clinical Outcomes and Indicators

NHS England

Reviewer	Comment	Response and/or amendment
1	How many of the characteristics intersect (i.e. Black, higher deprivation, young)?	Response There are relationships between the mother’s ethnicity, deprivation of residence, and age that we are now presenting in supplementary information. Amendment We have produced new tables in the supplementary information section (S8a – S8c) that present the intersection between the key pairs of demographic variables and we have discussed the results in supplementary information.
1	Are the trusts that place women in MCoC located in areas with higher social vulnerability?	Response We found no relationship between the deprivation of the area an NHS Trust serves and MCoC placement rates. Amendment We have produced a new scatterplot (figure S6) in the supplementary information section that shows MCoC placement against average deprivation by NHS Trust and this shows no relationship between the two variables at NHS Trust level. We have added the following to the Discussion section of the manuscript: “The relatively even distribution by demographic group, but wider variation by NHS Trust and Region in Supplementary Information suggests differing prioritisation between regions and organisations, but the analysis shows this is not related to deprivation of the areas they serve.”
1	How are women generally placed on MCoC? Are they assigned? Can women request it?	Response The NHS Trust assigns women based on where they have placed their MCoC teams. There is no mechanism for women to request placement. Amendment We have added the following statement to the introduction: “NHS Trusts decide where to place MCoC teams within their geographical area and consequently which women are placed on MCoC to best reduce inequalities.”

1	Would someone want to request it or are there drawbacks such as long waiting times or less desirable clinic locations?	Response There are no known drawbacks such as long waiting times or less desirable clinical locations. The introduction to the paper outlines some of the benefits identified in previous research and the discussion outlines benefits from this research. However, there is no mechanism for women to request placement as it is for NHS Trusts to determine who is placed.
1	Some more information might be helpful for non-English readers to understand the context of MCoC in the NHS.	Response The amendments that we have made in response to the above questions have provided some more information that will hopefully be helpful to non-English readers to understand the context of MCoC in the NHS.
1	I understand that you have beautiful granularity of data but I wonder if collapsing some of the categories might make the data more impactful? For example, I see that you have deciles of deprivation, it might make more meaningful and clinical sense to convert them to quartiles or quintiles. Similarly with race/ethnicity. I am not sure that the categories White and Black Caribbean vs White and Black African, while certainly meaningful to those individuals, might not be meaningfully different to readers or to clinicians caring for individuals from these groups. As I do not live in England, I do not know if this holds true in your context. The detailed tables could be included as supplementary with simpler tables included in the main portion of the article.	Response We thank reviewer 1 for raising this. We do not propose collapsing the categories in the table for the reasons set out below but have added some text to the Discussion section to justify the granularity of presentation. We agree fewer categories would enable results in the tables to be more immediately digestible. However, there are important distinctions in the English context between different ethnic groups for example Black African and Black Caribbean or Bangladeshi and Pakistani. Because we are very fortunate to have a large national dataset, the numbers in these categories, are large enough to produce meaningful results. The granular categories are what have been used as the independent variables in the logistic regression models for all women, so it would add a layer of complexity to the presentation and potentially lead to more tables if the odds ratios for certain demographic categories were distinct from the ratios and categories used in the models to test the hypotheses. Separately, we have broken down the hypothesis testing by broader ethnic and deprivation groups. The presentation of the intersections between the demographics in response to reviewer 1's very helpful suggestion also shows distinct patterns within an ethnic group. Amendment

		We have added some text to the discussion setting out the distinctions between different ethnic subgroups in the English context “Caution should be applied in generalising the findings on ethnicity to other countries. The local context on why and when immigration took place and levels of integration are likely to affect outcomes. There are likely to be important differences within broad ethnic groups based on this context and we have presented the most granular available ethnicity data in our descriptive tables, for example separating Pakistani and Bangladeshi ethnicities and separating Black African and Black Caribbean.” We hope that these additions will help provide the readers with more context on our approach and explain why we are reluctant to change the categories in the main tables. .
1	I would just like a bit more synthesis and also, discussion of the implications of the findings	Amendment We have expanded the discussion section to include more synthesis of the results and implications of the findings.
1	I am not sure why the Methods section is at the end of the manuscript rather than before Results	Amendment We have reordered the manuscript, so methods are before Results.
1	while the methods and analysis sections are extremely important to determine the rigor of a study, the multiple pages of Methods and Analysis are a bit long and too detailed for this descriptive study.	Amendment We have reduced the length and detail of the Methods and Analysis sections. As well as general streamlining, we have drawn together all the information around approaches for missing data and placing this in table S3 in supplementary information, which has also aimed to address some other reviewer feedback. We have made further reductions by moving some of the technical information to a technical specification section in the supplementary information after table S4. We have added some other specific aspects in response to other reviewer feedback and making explicit some points requested in the Communications Medicine reporting summary. We have looked to retain sufficient information to allow others broadly to replicate the study.

1	This is a confusing sentence as written, especially with inclusion of the word “delivery” which, for some readers, may mean “birth”: L48 In the United Kingdom (UK) National Health Service (NHS), the full range of maternity services are provided by NHS Trusts free at the point of delivery.	Amendment We have amended this sentence so it now reads: “In the United Kingdom (UK), the National Health Service (NHS), provides free of charge, comprehensive maternity services, which are available to all.”
1	In the first paragraph you twice state that midwifery continuity of care may have better outcomes, but you do not include any citations in either instance.	Amendment We have consolidated the first paragraph so this statement is made just once. We have added a citation to this statement (which had been cited later in the manuscript).
1	This feels a bit redundant as written: L62 NHS England is the national body that leads the NHS in England	Amendment We have amended the description of NHS England so it now states: “NHS England leads national healthcare policy”
1	Need a citation for these statements: L76 these rates have been found 77 to be up to twice as high as those for White women (CITE) and were still higher after adjusting for other 78 risk factors (CITE)	Amendment We have added citations directly against these statements, moving them from the end of the paragraph.
1	L87-90: You speak to negative experiences with maternity carers and their contribution to higher stillbirth rates. While this is likely true, you need to connect the dots a bit more here.	Amendment We have made this sentence slightly more circumspect, citing the precise evidence from the confidential inquiry: “among those women who experienced a stillbirth or neonatal death, significant issues in the quality of care were more common in Black than White women, including barriers to accessing specific care”
2	The title, "Closing the Inequalities Gap: Impact of Midwife Continuity of Carer on Birth Outcomes," appears broader than the study's scope, which focuses primarily on stillbirth and	Amendment We have amended the title so it now reads:

	breastfeeding initiation rates. Although the title highlights "birth outcomes," specifying the exact variables studied (e.g., stillbirth rates and first feed of breastmilk) would align it more closely with the manuscript's content. This change would provide clearer expectations to the reader and align better with the study's focus on minority and deprived populations.	"Closing the inequalities gap: impact of midwife continuity of carer on stillbirth rate and first feed of breast milk, an English population-based cohort study"
2	The section could benefit from reorganization to enhance clarity and flow. For instance, the context of perinatal care and the structure of MCoC in England should be introduced first, followed by a description of access disparities faced by minority groups and the overview of the study outcomes.	Amendment We have amended the introduction, so it reflects reviewer 2's recommended order, starting with the context of perinatal care and the structure of MCoC in England, followed by a description of access disparities faced by minority groups and the overview of the study outcomes.
2	The current structure of this section briefly mentions the two outcomes at first but without providing the rationale for their selection, making it unclear why these variables were analyzed together. Since they do not belong to the same category, this raises questions about the criteria used for selecting variables for analysis. Reorganizing the introduction would give readers essential context for understanding the choice of outcomes and independent variables examined.	Amendment We have provided clearer rationale for the selection of the two outcomes through an explicit statement against each one. The introduction now focuses in turn on the importance and target around reducing stillbirths in England with an explicit statement of its importance for the study, and the importance of first feed of breast milk, with an explicit statement of its importance for the study. By quoting the outcomes specifically in the title we hope this also adds clarity.
2	Lines 115-119 present the research objective described in the manuscript. The general aim stated at the beginning of the introduction appears somewhat misleading, thus perhaps it shouldn't be mentioned at this point and in	Amendment We have restructured the introduction so the hypotheses are placed together in once section and numbered towards the end of this section and are numbered. We have made amendments throughout the remainder of the manuscript to refer to these numbered hypotheses.

	such a general way. The study goals and hypotheses should also be placed together in one section along with their rationale at the end of the introduction section. It would be useful to pair the research objectives with the hypotheses (perhaps numbered) to make it easier to locate and reference them during the analysis of results.	
2	The line, "We avoided outcomes with poorer data quality or where clinical context of individual cases would be needed," (lines 118-119) is somewhat vague. Providing examples of outcomes that were excluded due to data limitations would clarify the study's design and limitations.	Response The data clearly highlighted whether a stillbirth or preterm birth had occurred or not, whereas other candidate outcomes such as postpartum tears, the absence of information on this could either mean one had not occurred or that the data were incomplete. Amendment We have added just above table S3 in the supplementary information some examples of what other candidate outcomes were and the reasons we did not include these in the study using the above text. We have strengthened the description of the positive reasons why we chose the outcomes we did investigate stillbirth rate and first feed of breast milk.
2	The Methods section, currently located after the Discussion, would ideally follow the Introduction to provide a logical flow.	Amendment We have reordered the manuscript, so Methods follows Introduction.
2	The variable "first feed of breastmilk" lacks clarity in terms of its measurement. Details on whether breastfeeding was directly observed or inferred based on database entries would strengthen this section.	Amendment We have added further text: "each NHS Trust is required specifically to record for each birth whether the baby's first feed was breast milk (specifying from mother or donor) or whether the first feed was not breast milk. It is envisaged they would do this from direct observation" to clarify this.
2	"NHS Trust" is mentioned in both the Introduction, Methods and Results, yet many	Amendment

	readers outside the UK may be unfamiliar with this term. Defining "NHS Trust" early on would aid comprehension.	We have added the following text to the first paragraph of the introduction "NHS Trusts are the organisations that deliver care within specific geographical areas, including services provided in hospitals and the community. "
2	Line 135: It might be helpful to repeat the full name, "Maternity Services Data Set," next to the abbreviation MSDS to remind readers what it stands for.	Amendment We have made this change to repeat the full name here.
2	Clarify whether data from only 121 Trusts were used, or whether this subset relates to the total number of Trusts in the UK, to enhance interpretation.	Response Data from all 124 NHS Trusts that provided publicly funded maternity services in England over the study period are included in the study. Figure 1 alone just included 121 Trusts as it set a restriction that there needed to be over 1,000 deliveries with known placement status in the period, so Figure 1 alone excludes the three Trusts with fewer than 1,000 deliveries with known placement status. Given that Reviewer 2 has helpfully highlighted this lack of clarity, and only three Trusts failed to make the inclusion criterion for Figure 1, which was established to avoid random fluctuation in smaller providers adversely affecting the interpretation, we have removed the exclusion criterion from Figure 1 so all 124 NHS Trusts are presented, rather than use words and complexity explaining this for such a small difference. Amendment The following text has been added to Methods: "All 124 NHS Trusts that provided NHS funded maternity services over the period had their data included." We have amended figure 1 so it includes all 124 NHS Trusts and the accompanying statement now reads: "Of the 124 Trusts in the study2..."
2	The discussion section covers the most important aspects of the results; however, expanding it could benefit readers.	We have added text to the Discussion relating to all reviewer 2's specific questions, and we have further expanded the discussion in response to reviewer 1 and reviewer 3's feedback.

2	Is the 20-30% placement of women in the MCoC what the authors expected?	Amendment We have added the below text to the discussion: “The overall proportion of 23% of women placed on MCoC broadly met expectations given the ambition outlined in 2021 for MCoC to be the default model of care but a further announcement in 2022 made the implementation timeline dependent on staff capacity.”
2	Discuss whether the results reveal any specific characteristics of the trusts with low vs. high MCoC placement rates, such as their location in more or less socially and economically deprived districts.	Amendment We have added the following text to the discussion: “The relatively even distribution by demographic group, but wider variation by NHS Trust and Region in Supplementary Information suggests differing prioritisation between regions and organisations, but the analysis shows this is not related to deprivation of the areas they serve.” To underpin this, we have produced a new scatterplot (figure S6) in the supplementary information section that shows MCoC placement against average deprivation of the women by NHS Trust. We have produced a new table (table s7) in the supplementary information section that shows MCoC placement by region of NHS Trust.
2	Discuss how we interpret the characteristics of the groups placed in the MCoC—is this distribution in line with expectations	Amendment We have added the following text to the discussion: “Lower variation in MCoC placement by other demographic groups suggests that within organisations there has generally been little targeting by demographic group, but this is likely to change under the core20PLUS5 policy.”
2	Discuss why the placement in MCoC had a relatively low impact on stillbirth rates, while the effect on the first breastmilk feed was more noticeable	Amendment We have added the following text to the discussion: “First feed of breast milk is a much more common outcome than stillbirth, so for first feed of breast milk a smaller change leads to much more significant results. However, of the two outcomes, stillbirth has a far greater impact so the fact that for Black women, MCoC placement has a significant impact on stillbirth outcomes is potentially a more important finding.”
2	Line 125: "If this, is the case..." should remove the comma.	Amendment We have removed this comma.

2	Line 133: "they had some worse birth outcomes..." could benefit from more formal wording, instead of the word 'some'.	Amendment We have replaced the word "some" so the sentence now reads: "It is possible that poorer interactions with maternity services over the course of a pregnancy would be a factor driving worse outcomes for higher risk groups".
2	Line 145: "not on placed on the MCoC" should be corrected to "not placed on the MCoC."	Response Due to the increase in the Discussion due to all reviewers' feedback, we have moved this paragraph to supplementary information where it accompanies the study flow diagram to keep the overall wordcount of the manuscript close to the guidelines. There are some high-level figures around the study composition in the methods section. Amendment We have removed the word "some" from this paragraph from within Supplementary Information.
2	395: "were selected" could be omitted	Amendment We have removed the words "were selected". There has been some other reordering as we have generally looked to reduce the word count in this section in response to reviewer 1's feedback.
2	Line 409: "was the total population used any aspects of the study." appears incomplete.	Amendment We have reworded this to state "This left a total study population of 922,149 pregnancies."
2	Line 468: "Table S5" should be updated to "figures S5a-S5e."	Amendment We have amended this reference to "Figures S12a-S12e", as due to reviewers' feedback there have been some additions and reordering to figures and charts within Supplementary Information.
3	Remove any mention of methods	Amendment We have removed any mention of methods from the introduction.
2	The results of the most and the least deprived quintile are also interesting, but can it be assumed that they cover the ethnicities already mentioned here? Nevertheless, since such	Response The most deprived quintile and the remaining least deprived 80% are a separate breakdown of the entire study population so do not include any further subdivision by ethnicity.

	grouping was included in the analyses, the results should be also interpreted	Amendment We have added the following commentary to the relevant part of Supplementary Information around the deprivation breakdown: “It also shows that those women living in the most deprived quintile who were placed on MCoC had significantly higher rates of first feed of breast milk compared with those in this quintile who were not placed. Significantly higher rates of first feed of breast milk among women placed on MCoC compared with those not placed were also observed for the women living in the least deprived 80% of the population.”
2	Why in this table some but not all results with sig <0.01 are in bold? It seems inconsistent, but perhaps there is other rule used?	Response Thank you to reviewer 2 for identifying this. We started highlighting significant results in bold but then stopped doing this but did not remove bold numbers from all of this table. Amendment We have removed all bold numbers to be consistent with other tables in the main manuscript and supplementary information.
2	I suggest revising this title, perhaps: summary of independent variables used in particular models? this table is very helpful in analysing the results, thus should be moved up if the analysis section is moved before the results section (which I strongly suggest)	Amendment We have amended the title to: “summary of independent variables used in each model” and moved it to earlier in the supplementary information section as Table S4.
2	If the methods section is moved after the introduction, then this figure might be moved to the first position in supplementary materials, as it provides useful information on the following steps of dataset is formed and managed	We have moved this figure so it is the first position in the supplementary materials as Figure S1.

3	For your introduction (i.e., the main), this could be set up clearer for readers. It is unclear from the introduction what the gap is trying to fill with this article. How is this paper different from the other continuity of carer articles out there including the systematic reviews on continuity of carer? I do appreciate the discussion of NHS England, but this should be discussed together. There should also be sentence about what is the aim of this study and your hypotheses which were described in line 115 and 120. I'm also confused why there was some discussion of the methods in this. The introduction could be setup in clearer way for readers.	Amendment We have restructured the introduction to reflect both reviewer 2's and reviewer 3's comments. We have added the below statement highlighting the gap that this paper is looking to fill: "Although systematic reviews have found benefits to MCoC, no research has taken place on real-world implementation on a national dataset, which is a significant gap as implementing an approach at scale across a population could have additional complexity that would influence the outcome, and larger national data allows testing on specific population subgroups." We have now referenced the aim of our study and hypotheses: "NHS England's access to rich, national data covering a woman's entire pregnancy and information on the type of maternity model of care a women received gave us a unique opportunity to investigate the effectiveness of the MCoC model on pregnancy outcomes. The aim of our study is to test:  • Hypothesis 1: women receiving MCoC had lower stillbirth rates and higher rates of a first feed of breast milk in England. • Hypothesis 2: women who had no antenatal appointments by 24 weeks have higher stillbirth rates and lower rates of a first feed of breast milk compared to the baseline group of women seen by 24 weeks but not placed on MCoC pathway." We have removed any reference to Methods from the introduction.
3	A study flow diagram would be helpful instead of the Table S1. I suggest putting this in the traditional study flow diagram.	Amendment We have replaced table S1 with a study flow diagram (figure S2).
3	In addition, why were there two different sample sizes?	Response Records for which the first feed of breast milk status was unknown were excluded from the component of the study looking at first feed of breastmilk but included in other aspects.

		Amendment We have looked to make this clear in the study flow diagram and in table S3 summarising the approach to missing and unknown data.
3	What are the “Trusts” referred to in line 150-151	Amendment We have added the following text to the first paragraph of the introduction “NHS Trusts are the organisations that deliver care within specific geographical areas, including services provided in hospitals and the community. “
3	Similarly with the discussion, there really is not a discussion of how their findings are similar or different from previous literature or what is known, which can be done with this study. I think this would make the paper stronger than discussing the results in further in-depth. In addition, I appreciate a discussion regarding the implications for practice and/or policy.	Amendment We have added to the discussion a comparison with the previous literature around the impact of MCoC on stillbirths and first feed of breast milk, and a comparison with the previous literature on the impact of late first antenatal appointments. We have added points around the impact on practice/or policy against the MCoC findings and against the late first appointment findings.
3		Amendment We have included a new table (table S3 in supplementary information) that presents by data item the number of missing values, our approach to missing values in that item and the rationale for this. We have run sensitivity analysis comparing the study’s approach (missing values as their own distinct category) with imputation and full case analysis for missing independent variables and presented this at table S10 in supplementary information. We have referenced the findings from this in the discussion section, where we have discussed the different possible approaches.
3	In the discussion, you mention that there was a missing data. What was the percentage of people who missing data? Did you do complete case analysis or imputation? In the methods section, you should include a description about	Response Four independent variables had missing or unknown values: ethnicity – 2.6% of records; parity 2.0% of records; previous caesarean 3.8% of records; and previous stillbirth - 1.8% of records. In addition, 0.7% of records had unknown deprivation of area due to no English address being

	how you handled this. I would like to understand more about the missing data because you mentioned these people had worse outcomes.	provided but this would include people who live outside England or have no address (homeless, recent immigrants) so is not necessarily missing. 8.0% of records had unknown first feed of breast milk. This is now all set out in table S3 of supplementary information. We gave each of the independent variables with missing/unknown values a distinct category of missing/unknown and fed this into the regression model. This is an approach that has been validated in other studies (e.g. Singh, Sato, Ohkuma: On Missingness Features in Machine Learning Models for Critical Care...). We performed sensitivity analysis presented in Table S10 in Supplementary Information that compares our approach with Imputation and Complete Case Analysis, which finds no major differences in the key findings regardless of which approach was taken. Amendment As well as adding tables S3 and S10 to Supplementary Information referenced above we have added a paragraph exploring the relative merits and impact of different approaches in the Discussion Section. We have also included the following text in the Methods section: “For some cases, first feed of breast milk status was unknown, and these cases were excluded from analysis of this outcome. Missing data for the independent variables were included as their own distinct category for that variable. Table S3 in Supplementary Information presents the detailed approach to missing data and its rationale.”
3	Discuss the generalizability of your study findings?	Amendment We have discussed the generalisability of the findings relating to MCoC and to the findings relating to late first appointments in the relevant sections of the Discussion.
3	Given the complex nature of race/ethnicity, I think there should be a discussion about this in your manuscript. Is it similar to the United States?	Response While there are similarities in some ethnic groups being disadvantaged and having worse outcomes, there are important distinctions based on the reason different ethnic groups live in a country.

		Amendment We have added the following paragraph to the discussion setting out the importance of local context in race/ethnicity: “Caution should be applied in generalising the findings on ethnicity to other countries. The local context on why and when immigration took place and levels of integration are likely to affect outcomes. There are likely to be important differences within broad ethnic groups based on this context and we have presented the most granular available ethnicity data in our descriptive tables, for example separating Pakistani and Bangladeshi ethnicities and separating Black African and Black Caribbean.”
3	I suggest using gender neutral terminology.	Response We agree that gender neutral terminology is important and where appropriate we have used this, but we do not think there is an unambiguous single word that would adequately distinguish the person who is pregnant from their partner/baby/clinician, so we have looked to address this via an upfront statement rather than amend the current wording. We could not find a Nature Portfolio style guide on this but are happy to defer to the editor’s view. The underlying dataset (MSDS) from which we have drawn the data refers to pregnant individuals as women. This may evolve in time and there may be future additions to the dataset to capture information on the pregnant individual’s gender identity, but under most circumstances we mirror the terminology in the dataset when reporting on it. Amendment We have added the following upfront statement to the report: “In this study, pregnant individuals are referred to as women as this is the terminology used to describe the participants in the national level, secondary data source that was utilized in this study. The authors acknowledge that this refers to the whole scope of individuals who may be pregnant, such as transgender and nonbinary individuals. “

		We have also added a sentence to the inclusion and ethics statement: “The maternity services dataset enabled granular breakdowns by ethnicity and the deprivation of the area in which the woman lived, as set out in the methods section, but does not currently capture information on gender identity so currently we are not able to monitor this important area.”
--	--	--